# Genomic insights into the role of *Salmonella* Typhi carriers in antimicrobial resistance and typhoid transmission in Urban Kenya

Susan M. Kavai[1]*, Winnie C. Mutai[2], Cecilia Mbae[1], Kelvin Kering[1], Ronald Ng'etich[1], Peter Muturi[1], Collins Kigen[3], Mike Mugo[1], Diana Imoli[1], Celestine Wairimu[1], Samuel Kariuki[1,4,5]

1 Centre for Microbiology Research, Kenya Medical Research Institute, Nairobi, Kenya, 2 Department of Medical Microbiology and Immunology, University of Nairobi, Nairobi, Kenya, 3 Walter Reed Army Institute of Research- Africa, Kericho, Kenya, 4 Wellcome Sanger Institute, Cambridge, United Kingdom, 5 Drugs for Neglected Diseases Initiative, Nairobi, Kenya

* ssnkavai8@gmail.com, skavai@kemri.go.ke

## Abstract

Typhoid fever cases and carriers can transmit *Salmonella enterica* serovar Typhi (*S.* Typhi) through fecal shedding. It remains unclear whether the *S.* Typhi shedding by carriers exhibits similar phenotypic and genotypic characteristics to those from acute cases. We investigated multidrug resistance in *S.* Typhi from individuals residing in urban informal settlements in Nairobi, Kenya. We recruited participants ≤ 65 years from six health facilities and tested for typhoid infection through blood and stool cultures. The *S.* Typhi culture-positive cases were treated and followed up after treatment, where index cases and their household contacts provided stool samples for culture. The susceptibility of all *S.* Typhi isolates was tested against 14 antibiotics using Kirby Bauer disc diffusion. Total deoxyribonucleic acid (DNA) was extracted from selected multi-drug resistant (MDR) *S.* Typhi for whole genome sequencing using Illumina Nextseq2000, and their genomes were analyzed on Pathogen-watch. Of the 115 *S.* Typhi isolates, 81/115 (70%) were from cases, while 34/115 (30%) were from carriers. *S.* Typhi resistance against ampicillin was observed in 32/81 (40%) and 11/34 (32%) of isolates from cases and carriers, respectively, while resistance against co-trimoxazole was observed in 34/81 (42%) and 10/34 (29%) of isolates from cases and carriers, respectively. In addition, resistance against chloramphenicol was observed in 30/81 (37%) and 10/34 (29%) in isolates from cases and carriers, respectively. Multidrug resistance was observed in 33% (38/115) of the *S.* Typhi isolates, with majority, 28/38 (74%) recovered from cases. A subset (22/38, 15 from cases and 7 from carriers) of the MDR isolates was randomly selected for sequencing. All the 22 *S.* Typhi belonged to genotype 4.3.1, with the majority 15/22 (68%) from genotype 4.3.1.2EA3. All these isolates carried the $bla_{TEM-1D}$, *catA1*, *dfrA7; sul1,* and *sul2* AMR genes. *GyrA* point mutations conferring reduced susceptibility to quinolones and fluoroquinolones were detected in 19/22 (86%) isolates, with the majority 15/22

**Data availability statement:** For relevant data please see the Supporting Information for GenBank accession numbers.

**Funding:** Research reported in this publication was supported by the National Institute of Allergy and Infectious Diseases of the National Institutes of Health under Award Number R01AI099525 to S.K. and the Wellcome Trust. The funders had no role in study design, data collection and analysis, decision to publish, or preparation of the manuscript.

**Competing interests:** The authors have declared that no competing interests exist.

(79%) occurring on codon 83. This study's findings highlight the plausibility of typhoid transmission within communities in disease endemic settings. Consequently, the study demonstrates the need for surveillance of antimicrobial resistance, antimicrobial stewardship, deployment of typhoid vaccine and improvement of water, hygiene and sanitation infrastructure in disease endemic settings.

## Introduction

*Salmonella* Typhi is a Gram-negative bacterial pathogen that causes Typhoid fever. Globally, typhoid is responsible for 21.7 million illnesses and 217 000 fatalities annually [1]. In low- and middle-income countries (LMICs), there are estimated 20.6 million cases of typhoid [2] with countries like Burkina Faso, Democratic Republic of the Congo (DRC), Ethiopia, Ghana, Madagascar, and Nigeria, recording more than 100 cases for every 100,000 person-years of observation [3]. Previous studies in Kenya have reported a 2–4% prevalence of typhoid with children 5–9 years old recording the highest overall incidence (crude, 208; adjusted, 359 per 100,000 pyo) [4,5].

Typhoid fever is a febrile systemic illness with multisystem signs and symptoms such as abdominal pain, anorexia, malaise, diarrhea, and fever [5]. Improper waste management, poor drainage systems, exposure to contaminated water and food, and poor hygiene conditions have all been related to an increased risk of infection in urban informal settlements in typhoid endemic regions [6]. The diagnosis of typhoid is challenging in low-resource settings because of the lack of appropriate laboratory equipment and unclear, non-specific clinical presentation of the disease [7,8,9]. The rapid diagnostic kits that are widely accessible are typically unreliable due to their lack of specificity [5]. The blood and bone marrow cultures are considered the gold standard,however, limited access and cost constraints limit their utilization in LMICs [7,10].

Another significant public health concern is the chronic carriage of typhoid, which occurs in 2–5% of individuals after initial resolution of the acute disease. The chronic *S*. Typhi carriers periodically excrete the organism in their stool or urine, which contributes to continued disease transmission [8,11,12]. The *S*. Typhi carriers disseminate the illness within their normal environment, leading to the spread of infections throughout the community and thereby sustaining an infectious reservoir. Previous reports on typhoid carriers showed paucity of data in relation to their AMR characterization especially on *S*. Typhi isolates from carriers found to be MDR [9].

Typhoid fever responds well to treatment with antibiotics; however, the rise of MDR *S*. Typhi strains overtime has led to less antibiotic options for treatment of complicated Typhoid [12]. Three decades ago, typhoid was treated using 1st line antibiotics such as ampicillin, chloramphenicol and co-trimoxazole, but overtime the emergence and persistence of multidrug resistance has led to their ineffectiveness [13]. This resulted in the use of fluoroquinolones such as ciprofloxacin or third-generation cephalosporins such as ceftriaxone to be considered as the antibiotics of choice for the treatment of typhoid in countries in Southeast Asia (SEA) and sub-Saharan (SSA) Africa, such as Kenya where typhoid fever is reported to be endemic [12,13,14].

Genotype 4.3.1 (H58) has been reported in previous studies in Asia and Africa as the most prevalent genotype associated with MDR *S.* Typhi [14]. The H58 was first reported in SEA 30 years ago and has since spread to SSA [14,15]. In Kenya, multiple studies have reported that MDR *S.* Typhi are predominantly H58 strains [4,5,16,17]. Antimicrobial genes associated with 1st line antibiotics such as $bla_{TEM}$, *catA1,* and *dfr1; sul1; sul2* have also been linked to MDR *S.* Typhi strains of H58 genotype [18,15,19]. *S.* Typhi mutations in DNA gyrase and topoisomerase iv enzymes in the quinolone resistance determining regions have been reported to alter the binding site of the fluoroquinolones. These mutations lead to increased resistance to fluoroquinolones, which is usually denoted by increased nalidixic acid resistance, and reduced susceptibility (intermediate) to ciprofloxacin.. Previous studies in SEA and SSA have reported a few *S.* Typhi isolates showing resistance to drugs like ceftriaxone [16]. The presence of MDR *S.* Typhi with additional resistance to fluoroquinolones and third-generation cephalosporin results in extensive drug resistance (XDR) and complicated treatment of typhoid caused by such strains [17,20–22].

Limited surveillance data exist on the antimicrobial resistance (AMR) phenotypes and genotypes associated with MDR *S.* Typhi among carriers, yet they could play an important role in the transmission of typhoid. This study, therefore, aimed to understand the genomic insights into the role of *S.* Typhi carriers in AMR and Typhoid Transmission in urban informal settlements.

## Materials and methods

### Study design

The study utilized a longitudinal study design to recruit participants and collect samples in various sites within Nairobi County from November 1, 2020, to May 17, 2021, and October 25, 2021, to November 30, 2022.

### Study sites and sampling setting

Six health facilities within the Mukuru informal settlements were selected for this study: five outpatient clinics (Medical Missionaries of Mary, Municipal City Council, Our Lady of Nazareth, Mukuru kwa Reuben-Nairobi metropolitan services, and Mukuru kwa Reuben) and one referral hospital (Mama Lucy Kibaki Hospital). These medical facilities serve the population residing within the Mukuru informal settlements. Typhoid cases were recruited from the healthcare facilities, while contacts (Individuals residing in the same household as the case) were recruited from households within the Mukuru area (Latitude: -1.3028 Longitude: 36.8843).

### Study Population

The study recruited symptomatic participants aged ≤65 years visiting the six healthcare facilities for treatment. Patients were included in the study if they presented with a history of fever lasting ≥ 3 days (an axillary temperature of ≥ 38 °C), with/without loose/liquid stools, and not on antibiotics at the time of presentation to the healthcare facilities. Participants who tested positive for *S.* Typhi were followed to their households, where household contacts without diarrhea in the past 30 days or typhoid symptoms at the time of visit were recruited. For the participants recruited from the health care facilities, individuals with non-typhoid-related symptoms or prior antibiotic use were excluded from the study. Additionally, household participants exhibiting typhoid-related symptoms or planning immediate travel outside Nairobi County were excluded.

### Typhoid cases follow-up and recruitment of asymptomatic household participants

Patients who tested positive for *S.* Typhi received treatment and were followed up after 14 days' post treatment. These index cases were requested to provide a stool/rectal sample to check for shedding. During follow-up, at least three asymptomatic household members living with the index case were invited to participate in the study after providing informed written and verbal consent. In instances where household contacts were less than three, the available ones were recruited. To

be included in the study, the household contacts needed not to have experienced diarrhea within the previous month and were able to provide stool/or rectal samples.

## Longitudinal Follow-up of typhoid cases and household contacts

The treated index cases and their household contacts were followed for three months to investigate convalescent shedding. In the first month, all participants underwent five follow-ups (day's 0- follow-up (F)1, 3-F2, 7-F3, 14-F4, 28-F5) to maximize on isolating *S*. Typhi with the high frequency of sample collection. Individuals with persistent shedding (stool/rectal cultures tested positive for *S*. Typhi) had monthly stool/rectal samples collected for up to the third month. However, the follow-up was stopped after three consecutive negative cultures for *S*. Typhi.

## Sample collection at the study sites and household levels

Upon presentation to the study sites, participants who met the inclusion criteria had their stool and blood samples taken. Participants who were unable to submit a stool sample had a rectal swab sample taken by the study team. During follow-ups, only stool/rectal samples were obtained from the treated index cases and the household participants. All the stool/rectal samples were transported in Cary Blair medium (Oxoid, Basingstoke, UK) in cold chain within 4 hours of collection.

At the study sites where only cases were recruited, aseptic venipuncture was used to obtain 1–3 ml and 8–10 ml of blood from children and adults, respectively. The blood was drawn using a sterile syringe, transferred to blood culture bottles, and shipped to Kenya Medical Research Institute (KEMRI) microbiology laboratory for processing in ambient temperatures.

## Microbiology laboratory processing of samples

**S. Typhi identification from blood samples.** A Bactec FX 40 Blood Culture System (BD, Franklin Lakes, New Jersey, USA) was used to incubate the blood cultures at 37 °C. Flagged positive blood cultures were removed from the bactec and subcultured onto MacConkey, chocolate, and blood agar plates (all from Oxoid, Basingstoke, UK). Suspect colonies were subjected to biochemical testing on analytical profile indexing (API20E) strips (API System, Montalieu Vercieu, France), and *S*. Typhi polyvalent O and monovalent antisera for further typing (9, d and vi) (Remel Europe Ltd).

**S. Typhi identification from stool and rectal samples.** Stool and rectal samples were placed into Selenite fecal broth media (Oxoid, Basingstoke, UK) for enrichment and incubated at 37 °C for 24 hours. Bacterial growth from the Selenite fecal broth media was plated onto MacConkey agar and Xylose lysine deoxycholate agar (both from Oxoid, Basingstoke, UK). Non-lactose fermenting colonies from MacConkey agar were biochemically confirmed using the API20E system (Montalieu Vercieu, France). Serological confirmation of *S.* Typhi was performed using polyvalent O and monovalent antisera (9, d, and vi) (Remel Europe Ltd).

## Antibiotic Susceptibility Testing (AST)

The Kirby Bauer disc diffusion method assessed the antibiotic susceptibility of *S*. Typhi isolates to 14 antibiotics which have activity against Gram-negative bacteria. Colonies were emulsified in normal saline to achieve a 0.5 MacFarland suspension. This suspension was evenly spread onto Mueller Hinton agar (Oxoid, Basingstoke, UK). Antibiotic discs dispensed on the agar included: amoxicillin-clavulanic acid (AMC, 10 (20/10) µg), ampicillin (AMP, 10 µg), azithromycin (AZM, 15 µg), cefotaxime (CTX, 30 µg), cefpodoxime (CPD, 30 µg), ceftriaxone (CRO, 30 µg), chloramphenicol (CHL, 30 µg), ciprofloxacin (CIP, five µg), gentamicin (GEN, 10 µg), kanamycin (KAN, 30 µg), nalidixic acid (NAL, 30 µg), co-trimoxazole (SXT, 25 µg), and tetracycline (TET, 30 µg). *Staphylococcus aureus* ATCC 25923 and *Escherichia coli* ATCC 25922 quality control organisms were tested alongside the test isolates. In addition, quality control measures (4mm media depth, 5 minutes wait after bacterial inoculation). The 2022 guidelines of the Clinical and Laboratory Standards Institute (CLSI M 100) were adhered to in interpreting the findings [23]. At a minimum, MDR *S.*Typhi were defined as

isolates showing resistance to ampicillin, chloramphenicol, and co-trimoxazole combined, which is the classical definition of MDR *S.* Typhi. A subset of these MDR *S.* Typhi was selected randomly for sequencing.

## Molecular laboratory processing methods

**DNA extraction and whole genome sequencing.** The Wizard Genomic DNA Extraction Kit (Promega, Wisconsin, USA) was used to extract genomic DNA from the selected MDR *S.* Typhi isolates in accordance with the manufacturer's instructions. A Nanodrop one spectrophotometer (ND-2000) (Thermo Fisher, Waltham, Massachusetts, USA) was used to quantify the DNA and assess the purity. Whole genome sequencing samples were prepped with the Illumina DNA Prep tag- mentation kit and unique dual indexes. Using a 300-cycle flow cell kit, sequencing was done on the Illumina NextSeq2000 platform to generate 2x150bp paired reads. Optimal base calling was supported by spiking 1%–2% PhiX control into the run. The NextSeq2000 was equipped with on-board analysis software called DRAGEN v3.10.12, which was used for read demultiplexing, read trimming, and run analytics.

**Genome assembly and bioinformatics analysis.** Raw reads were quality checked using FastQC v0.11.9 [24] and *de novo* assembled using Shovill v1.1.0 with adapter trimmer enabled. The quality of the assemblies was assessed with Quast, version 5.2.0, (https://github.com/tseemann/shovill). *S.* Typhi genomes were uploaded to Pathogen-watch (https://pathogen.watch/) for serotype confirmation and identification of AMR determinants. *Salmonella sp* serotypes were confirmed using the *Salmonella In Silico* Typing Resource (SISTR) database while Multi locus Sequence Typing (MLST) was ascertained using the Warwick EnteroBase (http://mlst.warwick.ac.uk/mlst/dbs/Sentericadatabase). The AMR genes present were identified using PAARSNP AMR - Library 90370 version 0.0.17 and *S.* Typhi genotypes confirmed using https://github.com/katholt/genotyphi database. Plasmid detection was performed using PlasmidFinder (https://cge.cbs.dtu.dk/services/PlasmidFinder/).

## Ethical considerations

Written informed consent was obtained from all the study participants. For participants aged ≤17 years informed consent was sought from their parents/guardians. Participants aged 12–17 years provided informed verbal assent in addition to their parents/guardians informed written consent. Ethical approval to conduct this study was sought from Kenya Medical Research Institute, Scientific and Ethics Review Unit (KEMRI/SERU/CMR/P00102/3781). Anonymity was maintained throughout the study period from participant recruitment to data analysis by using barcode labels and unique identification numbers.

## Data analysis

Descriptive statistics such as proportions and counts were used to analyze the socio-demographic characteristics of *S.* Typhi-positive individuals. These statistics were also used to compare the resistance profiles, distributions of AMR genes, and co-occurrence of the AMR genes among the study population. Statistical analysis was conducted using GraphPad version 8. Paired t-test was used to check for statistical significance of AMR patterns and genes present in cases and carriers. A confidence interval of 95% was used and $P < 0.05$ was considered as statistically significant. Pathogen-watch was used to prepare a phylogenetic tree to check for relatedness between the *S.* Typhi from cases and carriers. Visualization of the tree was done on micro-react an online tool.

## Results

### Study population characteristics

We analyzed 115 *S.* Typhi, 81/115 (70%) *S.* Typhi were from cases, and 34/115 (30%) were from carriers. Regarding age distributions, *S.* Typhi isolated from cases and carriers aged ≤20 years were most abundant at 52% & 65%, respectively

**Table 1. Socio-demographic characteristics of *S*. Typhi-positive individuals living in Mukuru, Nairobi, Kenya.**

| Variable | Cases, n = 81 | Carriers, n = 34 |
|---|---|---|
| **Age** | | |
| ≤20 years | 42 (52) | 12 (65) |
| 21-40 years | 32 (40) | 21(62) |
| 41-65 years | 7 (9) | 1(3) |
| **Gender** | | |
| Male | 48 (59) | 18 (53) |
| Female | 33 (41) | 16 (47) |
| **Specimen type** | | |
| Blood | 34 (42) | 0 (0) |
| Stool | 47(48) | 34(100) |
| **Study sites** | | |
| Mukuru area | 59 (73) | 22 (65) |
| Mama Lucy Kibaki | 22 (27) | 12 (35) |

*Footnote: Cases recruited at the outpatient health facilities and the level 5 hospital provided blood and stool while carriers recruited at households provided only stool samples.*

**Table 2. Comparisons of antimicrobial resistance profiles of *S*. Typhi among cases and carriers living in Mukuru, Nairobi, Kenya.**

| Antibiotics | Cases, N = 81, n (%) | Carriers, N = 34, n (%) |
|---|---|---|
| Ampicillin | 32 (40) | 11 (32) |
| Co-trimoxazole | 34 (42) | 10 (29) |
| Amoxicillin clavulanic acid | 1 (1.2) | 0 (0) |
| Ciprofloxacin | 1 (1.2) | 0 (0) |
| Nalidixic acid | 36 (44) | 15 (44) |
| Azithromycin | 5 (6) | 0 (0) |
| Gentamicin | 1 (1.2) | 0 (0) |
| Tetracycline | 3 (4) | 3 (9) |
| Chloramphenicol | 30 (37) | 10 (29) |

(Table 1). The gender distribution showed a predominance of *S*. Typhi in males compared to females among both cases (59%) and carriers (53%). *S*. Typhi were isolated more frequently from stool/rectal samples than from blood samples among cases (48%). Majority of the *S*. Typhi isolates were obtained from samples collected from Mukuru outpatient health facilities for cases (73%) and carriers (65%), respectively, Table 1.

### Antimicrobial resistance profiles of *S*. Typhi among cases and carriers

Multi-drug resistance was defined as *S*. Typhi combined resistance to ampicillin, co-trimoxazole, and chloramphenicol. Of the 81 cases, and 34 carriers, resistance against ampicillin was observed in 32/81 (40%) and 11/34 (32%) of *S*. Typhi from cases and carriers respectively. *S*. Typhi resistance to co-trimoxazole was seen in 34/81 (42%) of cases and 10/34 (29%) of carriers, while for chloramphenicol, *S*.Typhi resistance was present in 30/81 (37%) cases and 10/34 (29%) of carriers respectively.

Of the 115 *S*. Typhi strains analyzed for susceptibility to 14 antibiotics, 38/115 (33%) were found to be MDR; 28 from cases and 10 from carriers. Notably, *S*.Typhi resistance against nalidixic acid was observed in 37/81 (46%) of the

cases and 15/34 (44%) of the carriers, respectively, Table 2. None of the isolates in this study showed resistance to third-generation cephalosporins (ceftriaxone, cefotaxime, cefpodoxime). There was significant difference in the AMR profiles between cases and carriers, p value = 0.0178.

## Genomic features of the *S*. Typhi strains from cases and carriers

The *S*. Typhi isolates (n = 22) had genome sizes ranging from 4.7 to 4.9 Mb, with an average GC content of 52%. The average assembly N50 was 197 kb indicating substantial portion of the genome were assembled into relatively larger contigs and reflecting high-quality assemblies (S1 Table).

## Phylogenetic relatedness of MDR *S*. Typhi from cases and carriers

From the 22 sequenced isolates, all were confirmed to be *S*. Typhi (15 cases, 7 carriers) belonging to genotype 4.3.1. The 4.3.1 genotype was further subdivided into two lineages; 4.3.1.1 EA1 and 4.3.1.2 EA3, with lineage 2 sub lineage EA3 (4.3.1.2 EA3) being dominant. All *S*. Typhi isolates from cases and carriers showed close relatedness although they appeared to belong to different clades, Fig 1.

Notably, 4 *S*. Typhi isolates recovered from the same individual during acute disease (43198B and 43198S) and asymptomatic carriage (043198_F1 and 043198_F4) originated from the same branch forming a monophyletic group. The four isolates were also observed to be carrying *IncHI1A*/*IncHI1B (R27)* plasmids, Fig 1.

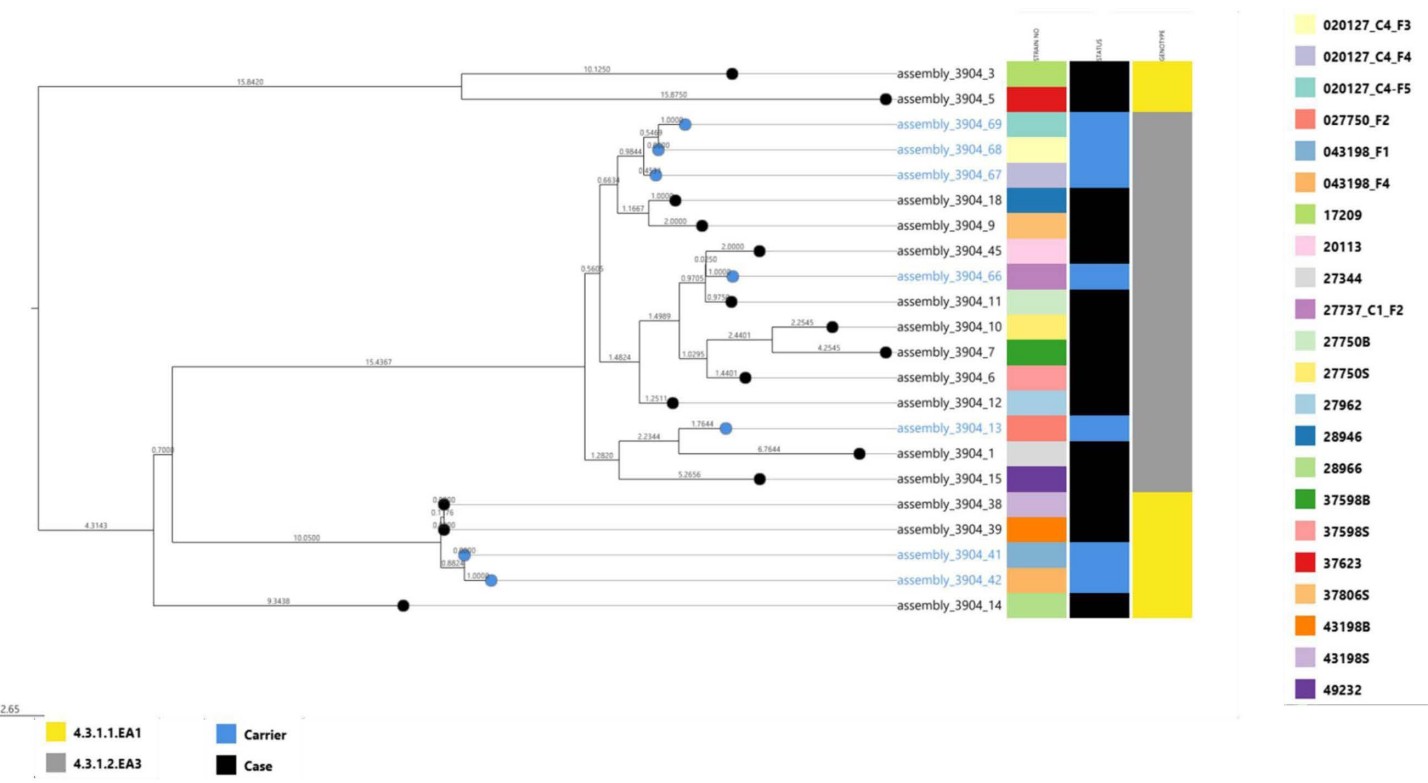

**Fig1. A phylogenetic tree showing the relatedness of *S*. Typhi from children living in Mukuru, Nairobi, Kenya.**

## Distribution of AMR genes among *S.* Typhi cases and carriers

All sequenced *S.* Typhi isolates AMR genes associated with first-line antibiotics. These included *bla*$_{TEM-1D}$, *catA1, dfrA7, sul1,* and *sul2*, which confer resistance to ampicillin, chloramphenicol, and co-trimoxazole, respectively. Each of these genes was observed in *S.* Typhi isolated from 15/22 (68%) cases and 7/22 (32%) carriers. Additionally, tetracycline (*tetA;tetB*) resistance genes were observed in 4 *S.* Typhi recovered from the same individual during acute disease and asymptomatic carriage, Fig 2. These tetracycline resistance genes were harboured in *IncHI1A/IncHI1B (R27)* plasmids. There was no statistical significance in the occurrence of AMR genes in *S.* Typhi from either cases or carriers (P = 0.0577), Fig 2.

## Point mutations in *S.* Typhi from the cases and carriers

Reduced susceptibility (intermediate) to fluoroquinolones (ciprofloxacin) was confirmed by the presence of *gyrA* mutations in 19 out of 22 (86%) of the sequenced MDR *S.* Typhi. These mutations occurred on codons 87 (*D87G*) and 83 (*S83Y*) in *S.* Typhi from both cases and carriers. At position *D87G*, it occurred in 2/22 (9%) of *S.* Typhi from cases and 2/22 (9%) of *S.*Typhi from carriers, respectively. For *S83Y*, the mutations occurred in 10/22 (45%) of the *S.* Typhi from cases and 5/22 (25%) of the *S.* Typhi from carriers.

## Shedding of *S.* Typhi among carriers

Seven *S.* Typhi isolates were recovered from 4 asymptomatic carriers (2 cases post-treatment and 2 contacts). Of the rounds of follow-ups, one contact was observed to be shedding MDR *S.* Typhi of the same lineage and sub-lineage (4.3.1.2 EA3) during three occasions of follow-ups (F3, F4, and F5).

In another household, a contact was observed to be shedding *S.* Typhi (4.3.1.2 EA3) only once during the second follow-up (F2). A case was observed to be shedding *S.* Typhi (4.3.1.1EA1) of the same lineage and sub lineage post-treatment during the first and fourth follow-ups (F1 and F4). A second case post-treatment was seen to be shedding *S.* Typhi (4.3.1.2 EA3) only once during the second follow-up (F2). In both cases, the S. Typhi strains recovered during acute disease were similar to those recovered during post treatment shedding.

The seven MDR *S.* Typhi were observed to have AMR genes (*bla*$_{TEM-1D}$, *catA1*, and *dfrA7; sul1; sul2*). The carrier reported to be shedding at F1 and F4 was also found to be shedding *S.* Typhi with *tetA* and *tetB* in addition to *bla*$_{TEM-1D}$,

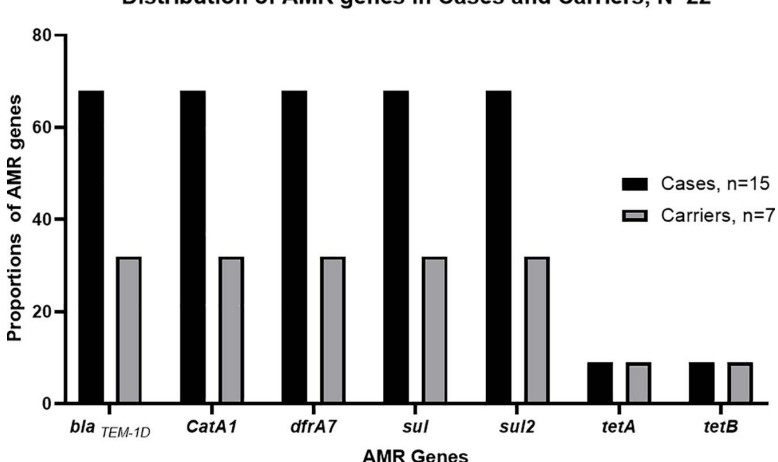

**Fig 2. Distribution of AMR genes in *S.* Typhi-positive individuals living in Mukuru, Nairobi, Kenya.**

*catA1*, and *dfrA7; sul1; sul2* AMR genes. Additionally, point mutations conferring reduced susceptibility to fluoroquinolones were detected in all 7 *S*. Typhi. This was indicated by the presence of *gyrA* gene mutations at codons 83 (*S83Y*) and 87 (*D87G*).

**Genotypes and co-occurrence of AMR genes in *S*. Typhi among cases and carriers**

The MDR *S*. Typhi isolated in the 3 years belonged to 2 genotype 4.3.1 lineages and sub lineages; 4.3.1.1.EA1 and 4.3.2 EA3. It was observed that in 2020, only 1 isolate from a case belonging to 4.3.1.1EA1 carried $bla_{TEM-1D}$/ *catA1*/ *dfrA7; sul1; sul2*, all associated with resistance to first line antibiotics. In 2021, it was observed that 17 *S*. Typhi belonging to 4.3.1.1.EA1 6 (35%) and 4.3.1.2EA3 11(65%) were isolated. Two isolates belonging to 4.3.1.1 EA1 from cases carried the $bla_{TEM-1D}$/ *catA1*/ *dfrA7; sul1; sul2* AMR genes while other 4 isolates, 2 from cases, and 2 carriers carried a combination of $bla_{TEM-1D}$/ *catA1*/ *dfrA7; sul1; sul2*/ *tetA; tetB*/ and a *gyrA* gene target mutation *(D87G)*. In the same year, 10 isolates belonging to 4.3.1.2EA3 were observed to be carrying a combination of $bla_{TEM-1D}$/ *catA1*/ *dfrA7; sul1; sul2*/ and a *gyrA* gene target mutation *(S83Y)*. In 2022, only 3 isolates belonging to genotype 4.3.1.2 EA3 were isolated and these were reported to be carrying a combination of $bla_{TEM-1D}$/ *catA1*/ *dfrA7; sul1; sul2*/ and a *gyrA* gene target mutation *(S83Y)*. From this data, it was conclusive that the point mutations associated with quinolone resistance, occurring on the *gyrA* on codon 87(D*87G)* were only occurring in MDR *S*. Typhi isolates belonging to 4.3.1.1 EA1 while for MDR *S*. Typhi belonging to 4.3.1.2 EA3, the point mutations occurring on *gyrA* were all on codon 83, (*S83Y*), Table 3.

## Discussion

Typhoid fever caused by MDR *S*. Typhi complicates patient management since it limits treatment options available in LMICs like Kenya where the disease is endemic. Since its emergence in SEA, *S*. Typhi, H58, has become the main lineage associated with typhoid in SSA and has persisted in this region. The presence of carriers shedding MDR *S*. Typhi harboring AMR genes and point mutations conferring reducing susceptibility to fluoroquinolones is worrisome since fewer antibiotics remain effective against typhoid fever [17,25].

This study revealed that *S*. Typhi isolates from index cases during acute disease and, post-treatment carriage and their household contacts exhibited resistance to similar classes of antibiotics. These isolates were resistant to ampicillin (40%& 32%,) co-trimoxazole (42% & 29%) and chloramphenicol (37% & 29%) from cases and carriers respectively. These findings, align with a previous study conducted in Kenya [12] which reported high antibiotic resistance to first-line antibiotics including ampicillin (64.9%), co-trimoxazole (59.6%), and chloramphenicol (51.7%), indicating the persistence of these resistant phenotypes in the community despite treatment interventions [12,16]. This highlights a critical issue in transmission dynamics likely driven by external factors including misuse of antibiotics through over-the-counter purchase, under dosages, and environmental contamination. Empirical treatment is also a potential contributor to AMR due to overtime

**Table 3.  Genotypes and co-occurrence of AMR genes carried by *S*. Typhi positive individuals living in Mukuru, Nairobi Kenya, n=22.**

| Year | Co-occurrence of AMR genes | Genotypes | Cases (n=15) | Carriers (n=7) |
|---|---|---|---|---|
| | | | n (%) | n (%) |
| 2020 | $bla_{TEM-1D}$/ *catA1*/ *dfrA7; sul1; sul2* | 4.3.1.1 EA1 | 1 (7) | 0 (0) |
| | $bla_{TEM-1D}$/ *catA1*/ *dfrA7; sul1; sul2*/ *gyrA S83Y* | 4.3.1.2 EA3 | 1 (7) | 0 (0) |
| 2021 | $bla_{TEM-1D}$/ *catA1*/ *dfrA7; sul1; sul2* | 4.3.1.1 EA1 | 2 (13) | 0 (0) |
| | $bla_{TEM-1D}$/ *catA1*/ *dfrA7; sul1; sul2*/ *tetA;tetB*/ *gyrA D87G* | 4.3.1.2 EA1 | 2 (13) | 2 (29) |
| | $bla_{TEM}$-1D/ *catA1*/ *dfrA7; sul1; sul2*/ *gyrA S83Y* | 4.3.1.2 EA3 | 9 (60) | 2 (29) |
| 2022 | $bla_{TEM-1D}$/ *catA1*/ *dfrA7; sul1; sul2*/ *gyrA S83Y* | 4.3.1.2 EA3 | 0 (0) | 3 (73) |

prescription of antibiotics without a confirmed diagnosis. The presence of these same variants in both *S*.Typhi from cases and carriers supports the assumption that AMR transmission can occur through shedding and re-transmission [19,25,26].

We observed reduced susceptibility to ciprofloxacin as denoted by high resistance to nalidixic acid by *S*. Typhi isolated from both cases (32%) and carriers (13%). In the context of fluoroquinolone resistance, mutations in the *gyrA* gene, lead to the amino acid changes *D83G* and *S87Y*. These mutants are not unique to Kenya as have previously been reported in Tanzania and Zanzibar [27]. In the Democratic Republic of Congo, a study reported that 30% of *S*. Typhi analyzed were MDR, of which 15% showed nalidixic acid resistance and decreased susceptibility to ciprofloxacin [28]. Ciprofloxacin is a current drug of choice for the treatment of typhoid disease in Kenya and therefore resistance has been selected independently multiple times and is ongoing. Extensive exposure to fluoroquinolones may have resulted in selective mutations in *gyrA*. The mutations observed may limit treatment choices [29,30].

Concordance was observed in the phenotypic and genotypic resistance with a high predominance observed for AMR genes associated with first-line antibiotics (*bla*$_{TEM-1D}$, *catA1, dfrA7; sul1; sul2*) in *S*. Typhi from both cases and carriers. The shedding of *S*. Typhi harboring these AMR genes by carriers enables these strains to persist and circulate within the population, as selective pressure created by antibiotic use favours their survival [28]. Previous studies on the prevalence of MDR *S*. Typhi found resistance genes similar to those observed in this study [12,19,30]. The widespread availability and misuse of these antibiotics without prescription foster the persistence of MDR variants harbouring these resistant genes [19,30]. The lack of significant difference in the occurrence of AMR genes in *S*. Typhi from either cases or carriers could be attributed to uniform exposure in this study setting.

MDR *S*. Typhi from cases and carriers were linked to a single lineage within the H58 clade, encompassing both the H58 lineage I sub-lineage East Africa I (4.3.1.1.EA1) and the H58 lineage II sub-lineage East Africa III (4.3.1.2.EA3) as previously observed in Kenya and Uganda [26,31]. This lineage is frequently associated with MDR phenotypes and reduced susceptibility to fluoroquinolones in Kenya. Previous studies trace the emergence of H58 to South Asia, where it gradually spread to sub-Saharan Africa, persisted and diversified alongside other variants for a long time [17,19,32]. From the phylogenetic tree, isolates from cases and carriers showed close relatedness. Notably 4 *S*. Typhi isolates from an individual during acute disease and in asymptomatic carriage showed origin from the same branch and very close relatedness forming a monophyletic group. This is an implication of transmission of *S*. Typhi strains with similar genetic makeup among the cases and carriers in this community.

Co-occurrence of other AMR genes among the MDR *S*. Typhi apart from those associated with first line antibiotics was observed. Presence of *tetA* and *tetB* resistance genes and single point mutations in the *gyrA* genes in the same isolates complicates treatment as fluoroquinolones are used when first line antibiotics fail. The co-occurrence of AMR genes in MDR *S*. Typhi observed did not show any discrimination towards a particular genotype or occurrence among cases and carriers. This implied that MDR *S*. Typhi are widespread and could complicate control efforts [17,19,29]. In this study we observed that the co-occurrence of multiple AMR genes with *tetA;tetB* was associated with presence of *(IncHI1A/IncHI1B (R27)* plasmids.

While the study identified key phenotype and genotypes linked to MDR *S*. Typhi, the study did not capture the full spectrum of all the MDR isolates and hence the findings may not be generalizable to all MDR *S*. Typhi strains in Mukuru informal settlement, and therefore future studies should expand the sequencing capabilities in order to provide a more comprehensive understanding of the genetic landscape of MDR *S*. Typhi in this setting.

In conclusion, our study reported presence of *S*. Typhi belonging to genotype 4.3.1 previously reported in Kenya, Eastern Africa and sub-Saharan Africa as the genotype responsible for multi drug resistance. Reduced susceptibility to ciprofloxacin indicated that treatment options for typhoid fever infections are limited [28,29,33]. It was evident that carriers and cases were shedding similar genotype that belonged to the same lineage and sub lineage. Presence of MDR *S*. Typhi from carriers circulating in the environment in this endemic setting poses risk of typhoid transmission to vulnerable individuals in this population.

Our study has a limitation in that a relatively small subset of *S*. Typhi isolates were sequenced. Subsequently, isolates from cases and carriers from the same household were not among the ones sequenced making it difficult to conclude if

they were shedding the exact same strain. Our study also carries a potential sampling bias due to recruitment only from Mukuru informal settlements in urban Kenya. This study reiterates the need for public health initiatives such as advocating for the appropriate use of antibiotics as well as the use of typhoid conjugate vaccine among vulnerable children as a control and preventive measure against typhoid [34,35,36].

## Supporting information

**S1 Table. Genomic features of S. Typhi from individuals living in Mukuru, Nairobi, Kenya.**
(DOCX)

## Acknowledgments

We wish to acknowledge the training and support from the University of Nairobi's Building Capacity for Writing Scientific Manuscripts (UANDISHI) Program at the Faculty of Health Sciences. We would also like to appreciate the study participants as well as the clinician and study staff who assisted in study participant recruitment, follow up and collection and processing of samples.

## Author contributions

**Conceptualization:** Susan M. Kavai, Winnie C. Mutai, Cecilia Mbae.

**Data curation:** Susan M. Kavai, Winnie C. Mutai, Celestine Wairimu, samuel kariuki.

**Formal analysis:** Susan M. Kavai, Winnie C. Mutai, Kelvin Kering, Ronald Ng'etich, Collins Kigen.

**Funding acquisition:** samuel kariuki.

**Investigation:** Susan M. Kavai, Peter Muturi, Mike Mugo, samuel kariuki.

**Methodology:** Susan M. Kavai, Ronald Ng'etich, Peter Muturi, Mike Mugo, Diana Imoli.

**Project administration:** Cecilia Mbae.

**Supervision:** Cecilia Mbae, samuel kariuki.

**Validation:** Kelvin Kering, Collins Kigen.

**Visualization:** Collins Kigen, Diana Imoli, Celestine Wairimu.

**Writing – original draft:** Susan M. Kavai.

**Writing – review & editing:** Susan M. Kavai, Winnie C. Mutai, Cecilia Mbae, Kelvin Kering, Ronald Ng'etich, Peter Muturi, Collins Kigen, Mike Mugo, Diana Imoli, Celestine Wairimu, samuel kariuki.

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
