## [Decision Letter · Decision Letter 0]

2 Jan 2025

PONE-D-24-48524Genomic analysis of Salmonella Typhi strains from typhoid patients and carriers reveals carriage as potentially important for community transmission of typhoid in KenyaPLOS ONE

Dear Dr. Kavai,

Thank you for submitting your manuscript to PLOS ONE. After careful consideration, we feel that it has merit but does not fully meet PLOS ONE’s publication criteria as it currently stands. Therefore, we invite you to submit a revised version of the manuscript that addresses the points raised during the review process.

We look forward to receiving your revised manuscript.

Kind regards,

Muhammad Qasim, Ph.D

Academic Editor

PLOS ONE

Journal Requirements:

“Funding

Research reported in this publication was partly supported by the National Institute of Allergy and Infectious Diseases of the National Institutes of Health under Award Number R01AI099525 to S.K. and the Wellcome Trust. 

Disclaimer. The content is solely the responsibility of the authors and does not necessarily represent the official views of the National Institutes of Health (NIH).”

3. Please note that funding information should not appear in the Acknowledgments section or other areas of your manuscript. We will only publish funding information present in the Funding Statement section of the online submission form. Please remove any funding-related text from the manuscript.

4. In the online submission form you indicate that your data is not available for proprietary reasons and have provided a contact point for accessing this data. Please note that your current contact point is a co-author on this manuscript. According to our Data Policy, the contact point must not be an author on the manuscript and must be an institutional contact, ideally not an individual. Please revise your data statement to a non-author institutional point of contact, such as a data access or ethics committee, and send this to us via return email. Please also include contact information for the third party organization, and please include the full citation of where the data can be found.

6. We note that Figure 1 in your submission contain map images which may be copyrighted. All PLOS content is published under the Creative Commons Attribution License (CC BY 4.0), which means that the manuscript, images, and Supporting Information files will be freely available online, and any third party is permitted to access, download, copy, distribute, and use these materials in any way, even commercially, with proper attribution. For these reasons, we cannot publish previously copyrighted maps or satellite images created using proprietary data, such as Google software (Google Maps, Street View, and Earth). For more information, see our copyright guidelines: http://journals.plos.org/plosone/s/licenses-and-copyright.

1) You may seek permission from the original copyright holder of Figure 1 to publish the content specifically under the CC BY 4.0 license.  

2) If you are unable to obtain permission from the original copyright holder to publish these figures under the CC BY 4.0 license or if the copyright holder’s requirements are incompatible with the CC BY 4.0 license, please either i) remove the figure or ii) supply a replacement figure that complies with the CC BY 4.0 license. Please check copyright information on all replacement figures and update the figure caption with source information. If applicable, please specify in the figure caption text when a figure is similar but not identical to the original image and is therefore for illustrative purposes only.

8. We note that there is identifying data in the Supporting Information file < Supp Data plos one.xlsx>. Due to the inclusion of these potentially identifying data, we have removed this file from your file inventory. Prior to sharing human research participant data, authors should consult with an ethics committee to ensure data are shared in accordance with participant consent and all applicable local laws.

-Location data

Please remove or anonymize all personal information (ID NO, Age in Months) ensure that the data shared are in accordance with participant consent, and re-upload a fully anonymized data set. Please note that spreadsheet columns with personal information must be removed and not hidden as all hidden columns will appear in the published file.

Reviewers' comments:

Reviewer's Responses to Questions

**Comments to the Author**

1. Is the manuscript technically sound, and do the data support the conclusions?

Reviewer #1: No

Reviewer #2: Yes

2. Has the statistical analysis been performed appropriately and rigorously? 

Reviewer #1: No

Reviewer #2: Yes

3. Have the authors made all data underlying the findings in their manuscript fully available?

Reviewer #1: Yes

Reviewer #2: Yes

4. Is the manuscript presented in an intelligible fashion and written in standard English?

Reviewer #1: No

Reviewer #2: Yes

5. Review Comments to the Author

Reviewer #1: This study followed patients from whom S. typhi was recovered and their household contacts to see who had the bacteria and for how long. It divides the subjects into cases and carriers, but unfortunately never defines the carriers or says how long they were positive for S. typhi in their stools or rectal swabs. Overall, the study might be interesting, but the presentation seems like a first draft, making it hard to decipher the findings. There are too many problems and gaffs to list them all, so just a few examples will be noted below. The authors should work further on the manuscript so that it concisely recounts the findings, with some good copy editing before submitting again. Salmonella typhi should written in italics with the t in lower case: S. typhi.

It is hard to tell from the abstract what were the main findings. It is a list of findings with too many details that don’t fit into a coherent message. It talks about transmission from carriers, but transmission is not discussed in the results section of the manuscript. If most of the isolates in the community are similar, how can it be clear that transmission comes from within the same household. If the strains were genome sequenced, the genomic distance between the isolates, or the presence of the same plasmids could be used to imply transmission chains, but this was not done. There isn’t even any data on how many positives were in the different households, or how they might have transmitted their infections.

Line 66 and others: in “Globally, the TF…”, the word “the” is unnecessary.

The introduction is too long and goes into details about gallbladder colonization and other topics that are unnecessary.

Line 125 What is the definition of a carrier? Any asymptomatic household contact with S. typhi in the stools or rectal swab? How long did they carry, how many per household. Were the isolates from the cases and their household contacts always exactly the same?

Figure 1 – the schematic map on the right is impossible to read because the font is too small

Line 134 What is meant by “with/out”

Line 242. Presumably they mean that P values < 0.05 were taken as significant, but this is not stated.

Line 245 What is meant by the sentence, “S. Typhi isolated from both cases

and carriers aged ≥20 years were most prevalent at 52% &65% cases and carriers

respectively.”?

Line 251 THE majority

Lines 253 – 4. The meaning is not clear.

Line 262. 11 (10%) of cases. The “of” is not needed. And it is 10% of what? 11/??, same in the next line: 34/?? This is a recurrent problem: it is not clear what populations these are percentages of.

Line 267 Of what are 15 isolates with nalidixic acid resistance 13%? This is a problem throughout the manuscript

Figure 2 The overall number of cases and controls should be shown in the legend

Lines 283-5 Lineage 1 is not discussed previously and only mentioned in the discussion line 392-3, so in this location in the results it is unknown what is being referenced.

Line 302-3 are unclear

Figure 3 is odd and the format makes it difficult to understand, in addition, the in the purple boxes the background is too dark for the dark letters to be legible.

Line 311 “2(95) in carriers”. What is 95? The “in” is unnecessary.

The discussion is much too long and rambling. Fluoroquinolone resistance could be treated much more concisely

Lines 411-13. If the AMR genes showed no discrimination, how could transmission be identified?

The study should be worth publishing, but much more work is needed on analysing the data and presenting it in a concise, comprehensible fashion. An interesting question, not addressed, is why there was more antibiotic resistance in isolates from cases than from carriers, and if there was a difference, doesn’t that complicate assertions of transmission between cases and carriers. Maybe not, but no data for this is shown concerning transmission.

Reviewer #2: The manuscript "Genomic analysis of Salmonella Typhi strains from typhoid patients and carriers reveals carriage as potentially important for community transmission of typhoid in Kenya" has been thoroughly examined. The study offers significant discoveries regarding the genomic characterization of Salmonella Typhi and its connection to the spread of typhoid and antibiotic resistance. However, the manuscript needs to be revised. Here are some specific areas that require improvement:

Title and Abstract

• Take into consideration making the title more focused and specific. For example: "Genomic Insights into the Role of Salmonella Typhi Carriers in Antimicrobial Resistance and Typhoid Transmission in Urban Kenya."

• The focus on carriers and AMR profiles in the study should be clearly reflected in the title.

• The primary findings, including the frequency of genotypes and associated AMR patterns, should be outlined in the abstract. Public health recommendations should be included at the end.

• Give a one-sentence explanation of the methodology, highlighting AMR characterization and whole genome sequencing.

• Do not repeat information (for example, resistance to first-line antibiotics is mentioned twice).

Introduction

• More quantitative information regarding the prevalence of typhoid fever in Kenya and its regional and worldwide distribution should be provided.

• Explain briefly why disease transmission is most prevalent in urban informal communities.

• Draw attention to inadequacies in earlier research, especially with relation to carriers' involvement in AMR transmission.

• Although the novelty and rationale are not stated clearly, describe how the study fills in knowledge gaps about the role of carriers in AMR.

• The introduction briefly mentions AMR-associated genotypes. Expound on why genotype 4.3.1 (H58) is significant in the Kenyan context.

Materials and Methods

• Although the longitudinal design is obvious, the rationale behind the frequency of sample collection and follow-up periods should be given.

• Explain the selection of medical facilities for hiring.

• The Cary Blair medium is suitable for transporting S. Typhi, but to maintain sample integrity, storage conditions and maximum transit times should be considered.

• Discuss whether AST made use of quality control measures other than reference strains?

• Justify the inclusion of antibiotics, especially third-generation cephalosporins.

• Protocols for DNA extraction and sequencing are described in depth. But make it clear: (i) Read quality and depth coverage thresholds, (ii) How mistakes were reduced during assembly, (iii) MDR isolate selection criteria for sequencing.

• Discuss statistical techniques for comparing the prevalence of AMR in carriers and cases. Indicate the program used and any modifications made to account for possible confounders.

Results

• Use the proper statistical metrics (such as p-values and confidence intervals) to quantify the observed variations in resistance between cases and carriers.

• Discuss genomic completeness criteria (such as N50) and the cutoff points for evaluating the quality of an assembly.

• To confirm their authenticity, compare genome sizes and GC content to previously published data for S. Typhi.

• Emphasize the importance of genotypes in connection to AMR (for example, 4.3.1.1 EA1 vs. 4.3.1.2 EA3).

• Discuss why blaTEM-1D, catA1, and other AMR genes co-occur with tetA and tetB.

• The table of genetic characteristics provides useful information. For a more thorough perspective, include summary data (such as the mean genome size).

• Provide a summary table that compares the prevalence of AMR statistically between carriers and cases.

• It is important to interpret certain statistical findings, such as the lack of significance for the prevalence of the AMR gene between cases and carriers (e.g., limited sample size or uniform exposure).

Discussion

• Provide more details on novel discoveries, like genotypic variants unique to the Kenyan context, to support comparisons to earlier research.

• Explain how AMR trends may affect empirical treatment plans in Kenya?

• Give specific suggestions for public health initiatives, including immunization drives or more stringent antibiotic stewardship in cities.

• Although the discussion of bile-induced transcriptional alterations in carriers is intriguing, it may be extended to clarify how this supports the persistence of AMR.

• Add limitations such as the relatively small subset of sequenced isolates and potential sampling bias due to recruitment only from Mukuru.

6. PLOS authors have the option to publish the peer review history of their article (what does this mean? ). If published, this will include your full peer review and any attached files.

**Do you want your identity to be public for this peer review?** For information about this choice, including consent withdrawal, please see our Privacy Policy .

Reviewer #1: **Yes: ** Howard E. Takiff

Reviewer #2: No

---

## [Author Response · Author response to Decision Letter 0]

6 Mar 2025

Editors' comments and my responses

Comment: Please note that funding information should not appear in the Acknowledgments section or other areas of your manuscript. We will only publish funding information present in the Funding Statement section of the online submission form. Please remove any funding-related text from the manuscript.

Response: I have removed the funding information from the manuscript. It now only appears on the online submission system.

2. We note that Figure 1 in your submission contain map images which may be copyrighted. All PLOS content is published under the Creative Commons Attribution License (CC BY 4.0), which means that the manuscript, images, and Supporting Information files will be freely available online, and any third party is permitted to access, download, copy, distribute, and use these materials in any way, even commercially, with proper attribution. For these reasons, we cannot publish previously copyrighted maps or satellite images created using proprietary data, such as Google software (Google Maps, Street View, and Earth). For more information, see our copyright guidelines: http://journals.plos.org/plosone/s/licenses-and-copyright.

1) You may seek permission from the original copyright holder of Figure 1 to publish the content specifically under the CC BY 4.0 license.

2) If you are unable to obtain permission from the original copyright holder to publish these figures under the CC BY 4.0 license or if the copyright holder’s requirements are incompatible with the CC BY 4.0 license, please either i) remove the figure or ii) supply a replacement figure that complies with the CC BY 4.0 license. Please check copyright information on all replacement figures and update the figure caption with source information. If applicable, please specify in the figure caption text when a figure is similar but not identical to the original image and is therefore for illustrative purposes only.

Response: The map image that previously appeared in the methodology section was removed from the manuscript and does not appear in the revised manuscript.

3. Please ensure that you refer to Figure 2 in your text as, if accepted, production will need this reference to link the reader to the figure.

Response: Figure 2 has now been mentioned in the manuscript on lines 310 & 313.

---

## [Editor Report · Decision Letter 1]

12 Mar 2025

Genomic Insights into the Role of Salmonella Typhi Carriers in Antimicrobial Resistance and Typhoid Transmission in Urban Kenya

PONE-D-24-48524R1

Dear Dr. Kavai,

We’re pleased to inform you that your manuscript has been judged scientifically suitable for publication and will be formally accepted for publication once it meets all outstanding technical requirements.

Kind regards,

Muhammad Qasim, Ph.D

Academic Editor

PLOS ONE
---

## [Editor Report · Acceptance letter]

PONE-D-24-48524R1

PLOS ONE

Dear Dr. Kavai,

I'm pleased to inform you that your manuscript has been deemed suitable for publication in PLOS ONE. Congratulations! Your manuscript is now being handed over to our production team.

Kind regards,

on behalf of

Dr. Muhammad Qasim

Academic Editor

PLOS ONE